# Identification of Novel Loci and Candidate Genes for Cucumber Downy Mildew Resistance Using GWAS

**DOI:** 10.3390/plants9121659

**Published:** 2020-11-27

**Authors:** Xiaoping Liu, Hongwei Lu, Panna Liu, Han Miao, Yuling Bai, Xingfang Gu, Shengping Zhang

**Affiliations:** 1Institute of Vegetables and Flowers, Chinese Academy of Agricultural Sciences, Beijing 100081, China; 82101181120@caas.cn (X.L.); lhw516@126.com (H.L.); liupn88@126.com (P.L.); miaohan@caas.cn (H.M.); 2Plant Breeding, Wageningen University & Research, Droevendaalsesteeg 1, 6708 PB Wageningen, The Netherlands

**Keywords:** allele-mining, downy mildew resistance, GWAS, polygenic, recessive

## Abstract

Downy mildew (DM) is one of the most serious diseases in cucumber. Multiple quantitative trait loci (QTLs) for DM resistance have been detected in a limited number of cucumber accessions. In this study we applied genome-wide association analysis (GWAS) to detected genetic loci for DM resistance in a core germplasm (CG) of cucumber lines that represent diverse origins and ecotypes. Phenotypic data on responses to DM infection were collected in four field trials across three years, 2014, 2015, and 2016. With the resequencing data of these CG lines, GWAS for DM resistance was performed and detected 18 loci that were distributed on all the seven cucumber chromosomes. Of these 18 loci, only six (*dmG1.4, dmG4.1, dmG4.3, dmG5.2, dmG7.1*, and *dmG7.2*) were detected in two experiments, and were considered as loci with a stable effect on DM resistance. Further, 16 out of the 18 loci colocalized with the QTLs that were reported in previous studies and two loci, *dmG2.1* and *dmG7.1*, were novel ones identified only in this study. Based on the annotation of homologous genes in *Arabidopsis* and pairwise LD correlation analysis, several candidate genes were identified as potential causal genes underlying the stable and novel loci, including *Csa1G575030* for *dmG1.4*, *Csa2G060360* for *dmG2.1*, *Csa4G064680* for *dmG4.1*, *Csa5G606470* for *dmG5.2*, and *Csa7G004020* for *dmG7.1*. This study shows that the CG germplasm is a very valuable resource carrying known and novel QTLs for DM resistance. The potential of using these CG lines for future allele-mining of candidate genes was discussed in the context of breeding cucumber with resistance to DM.

## 1. Introduction

Cucumber (*Cucumis sativus* L.) is an economically important vegetable crop produced all over the world. Downy mildew (DM), caused by the obligate biotrophic oomycete *Pseudoperonospora cubensis* ((Berkeley and Curtis) Rostovzev), is one of the most serious diseases threatening cucumber production [1,2]. The symptoms of cucumber DM mainly appear on the upper side of infected leaves starting with chlorotic (yellow) lesions that are restricted by major leaf veins. Symptom development is sensitive to environmental conditions and in favor of wet and humid conditions, since the pathogen needs moisture for germination and starting of a new infection. Within days following the infection under appropriate environmental conditions, chlorotic lesions become necrotic and sporulation (with a brownish gray downy appearance) may be seen on the lower side of infected leaves. With the disease growth, infected leaves will eventually collapse, leading to losses of cucumber production due to lack of photosynthetic leaves.

The cultivated cucumber was domesticated from a wild progenitor *C. sativus* var. *hardwickii* by Indian people at least 3000 years ago and brought to China ca. 2000 years ago [3]. Today, over 75% of all cucumber produced worldwide are grown in China (FAOSTAT, 2017) (http://www.fao.org/faostat). In addition to the cultivated cumber, another type of variety, *C. sativus* var. *xishuangbannanesis*, is recognized and cultivated in Southern China [4]. In Indian and Chinese cucumber accessions, resistance to DM has been identified and introgressed into commercial cultivars. From Chinese accessions, PR lines with DM resistance were generated at the Puerto Rico Agricultural Research station in the 1930s and 1940s [5]. The first DM resistant cultivar, Palmetto, derived from one PR line (PR40) was released in 1948 [6]. Two years later, the DM resistance of Palmetto was overcome [7] and another source of resistance was identified in the Indian accession PI 197087 [8]. The DM resistance in PI 197087 was shown to be controlled by a recessive gene *dm-1*, which is a mutant allele of the *Staygreen* (*Sgr*) gene [9]. The *dm-1* gene is located on chromosome 5 and had been used to control DM for nearly 50 years before it was overcome in 2004 by new DM strains [10]. After the breaking of the *dm-1* resistance, many other quantitative trait loci (QTLs) for polygenic DM resistance have been reported in different cucumber accessions and these QTLs are located over all cucumber chromosomes (Table 1). Very recently, it was shown that the QTL DM4.1 identified in PI 197088 contains three subQTLs, with each contributing to chlorosis, sporulation, and necrosis, respectively. One cucumber receptor-like kinase (RLK) gene *CsLRK10L2* is potentially the casual gene for subQTL DM4.1.2 contributing to sporulation [11]. Further, a loss-of-function mutation in the cucumber *Amino Acid Permease 2A* (*CsAAP2A* gene) underlies the resistance conferred by subQTL4.1.3 through limiting the DM-induced chlorosis [12]. The QTL regions on chromosomes 4 and 5 have been detected in different studies/accessions, indicating that these are major and stable QTLs contributing to DM resistance. It remains to be seen whether the overlapping QTL regions carry the same causal genes or different alleles of the same underlying genes, for example the *Sgr, CsLRK10L2,* and *CsAAP2A* genes.

Clearly, the study on DM resistance in cucumber is limited to a few genotypes listed in Table 1, and the accession PI 197088 has been the most studied and used one in breeding programs. In this study, we uncovered the genetic architecture on DM resistance in a core germplasm (CG) of cucumber re-sequenced lines [13], which represent cucumber genotypes with diverse origins and ecotypes.

## 2. Results

### 2.1. Genetic Diversity of DM Resistance in the CG Germplasm

Based on natural infection, the CG lines were scored for DM resistance in the spring of 2014 (dm_2014S), the autumn of 2014 (dm_2014A), the autumn of 2015 (dm_2015A), and the spring of 2016 (dm_2016S), respectively (Appendix A, Figure 1). The mean DI of dm_2014A and dm_2015A was higher than that of dm_2014S and dm_2016S (Figure 2a), indicating that DM was more severe in the autumn. In total, four cucumber ecotypes were evaluated, including the East Asian type (*n* = 37 CG lines), the Eurasian type (*n* = 30), the Indian type (*n* = 22), and the Xishuangbanna type (*n* = 8) (Appendix A). A similar mean DI was found for all market types except for the Xishuangbanna type that had a lowest mean DI in the spring of 2014 (Figure 2b). However, it was difficult to conclude that the Xishuangbanna type was more resistant since only eight CG lines were included and DI data were missing for most of them in two experiments (Appendix A).

Based on the average DI per line of the four experiments, these CG lines could be grouped into three clusters—1: Susceptible, 2: Intermediate resistant, and 3: Resistant (Figure 1). Each cluster contains different ecotypes (Appendix A). However, disease scores of some lines differed greatly among experiments. For example, the CG18 and CG14 line in the resistant cluster were susceptible in the spring of 2014 (dm_2014S) (Figure 1). This could be due to the fact that the naturally occurring DM infection was possibly not evenly distributed. Thus, we combined all phenotypic data of the four experiments to differentiate resistant and susceptible lines. A CG line was considered highly susceptible (HS) if it had a DI above 50 in at least one of the four experiments, or highly resistant (HR) if it had a DI smaller than 35 in all the experiments (Figure 1). From the tested 97 CG lines, we could define 18 HR and 22 HS lines, which are presented by different ecotypes (Appendix A). Further, a normal DI distribution was found for three of the four experiments except for the spring of 2016, which suggested that DM resistance in this CG population was controlled by multiple QTLs (Appendix A). Significant correlations were found among DIs of three experiments, dm_2014S, dm_2014A, and dm_2015A (Appendix A). Since there was no good correlation of dm_2016S with other years, DI data of 2014 and 2015 were used for further analysis.

### 2.2. Genome-Associated Analysis of DM Resistance

Using the DI data of the three experiments in 2014 and 2015, genome-wide association analysis (GWAS) was performed and 18 genetic loci were detected contributing to DM resistance. These 18 loci were distributed over the seven chromosomes (Figure 3). Of the 18 loci, only six loci (*dmG1.4, dmG4.1, dmG4.3, dmG5.2, dmG7.*1m and *dmG7.2*) were repeatedly detected in two experiments over two years, which were considered loci with a stable effect on DM resistance. By comparing the SNP (single nucleotide polymorphism) positions of these 18 loci with QTLs reported in previous studies (Appendix A), it was evident that 16 of the 18 loci of this study colocalized with the known ones (Figure 4). Only two loci, *dmG2.1* and *dmG7.1,* had not been reported before and were therefore considered as novel loci for DM resistance.

### 2.3. Candidate Gene Analysis for the Novel and Stable Loci

In order to identify potential candidate genes in novel and stable loci (detected in more than one experiment), the 50-kb region around the peak SNPs was used for further analysis with the cucurbit genome (http://cucurbitgenomics.org/). The choice of the 50-kb region was based on the fact that LD decay was less than 56 kb in most material in cucumber [13].

In the 50-kb region of the novel locus *dmG2.1*, six genes were present (Figure 5a,b): *Csa2G059820* (encoding a pentatricopeptide repeat-containing protein), *Csa2G059830* (a homogentisate phytyltransferase), *Csa2G059840* (a lysine-tRNA ligase), *Csa2G060350* (a transcription factor/transcription regulator), *Csa2G060360* (an unknown protein), and *Csa2G060370* (a Fe/S biogenesis protein). Only SNPs in the *Csa2G059830* gene showed a significant DI difference between haplotypes of the defined HS and HR lines (using DI data of three experiments: Spring in 2014, autumn in 2014, and autumn in 2015) (Figure 5c). In the HR and HS lines, 18 of the 22 HS lines carried the ATC haplotype, while 11 of the 18 HR lines had the GCT haplotype (Appendix A, Figure 5b). According to the transcriptomic data of PRJNA285071 (http://cucurbitgenomics.org/rnaseq/cu/19), the expression of *Csa2G059830* was higher in the susceptible material (Vlaspik) compared with the resistant material (PI 197088). Further, the expression was induced in both Vlaspik and PI 197088 upon the DM infection (Figure 5d).

For the novel locus *dmG7.1*, four genes were predicted in the 50-kb region (Figure 6b). All four genes encoded a protein related to the isoflavone reductase. Among the SNPs in these four genes, six SNPs were found in the intron region of *Csa7G002520* (Figure 6c) and showed significance on DIs between haplotypes of the defined HR and HS lines (data of two experiments: Spring in 2014 and autumn in 2015). In the HR and HS lines, 15 out of the 18 HR lines carried the TCTAAC haplotype, while 10 out of the 22 HS lines had the GTGGG haplotype (Appendix A) (Figure 6b). The expression level of this gene was comparable between Vlaspik and PI 197088 until 3 days post-inoculation (dpi) with DM (Figure 6d), while the expression was induced in both genotypes starting from 2 dpi and the induction was stronger in PI 197088 than Vlaspik.

For the stable locus *dmG1.4*, the candidate region (Chr.1: 21,711–21,781 kb) was analyzed by pairwise LD correlations leading to a region from 21,753,524 to 21,774,330 bp by LD block (Figure 7a). Five candidate genes were located in this region (Figure 7b). However, none of the SNPs in these genes showed a significant DI difference between haplotypes of the defined HS and HR lines. Nevertheless, the *Csa1G575010* gene (encoding a pectinesterase) was interesting since SNPs were located in the CDS (coding sequence) region of this gene that led to amino acid variation (Figure 7c). In the HR and HS lines, all the 18 HR lines carried the TAGTTT haplotype, while only 6 out of the 22 HS lines had the alternate GGCACC haplotype (Appendix A). Although the DI of the lines carrying the TAGTTT haplotype was lower than the alternate haplotype, the difference was not significant (using data of three experiments: Spring in 2014, autumn in 2014, and autumn in 2015) (Figure 7d). Interestingly, it was highly induced starting from 2dpi in the resistant genotype PI 197088 in contrast to almost no detected expression in the susceptible genotype of Vlaspik (Figure 7e). Possibly, the expression of this gene might play a role in DM resistance.

For the locus *dmG5.2*, the 50-kb region around the peak SNPs was further analyzed and a candidate region from 22,883 to 22,913 kb (~30kb) (Figure 8a) was estimated using pairwise LD correlations (r2 ≥ 0.6). In this region, five annotated genes were identified (Figure 8b): *Csa5G606470* (a WRKY transcription factor 2–4), *Csa5G606480* (an eukaryotic translation initiation factor 5A), *Csa5G606490* (a cell division topological specificity factor), *Csa5G606500* (a putative uncharacterized protein RAF9-1), and *Csa5G606510* (an unknown protein). One SNP_2531403 was located in the CDS of *Csa5G606470.* On this site, six of the 18 HR lines was G (Pro), and all the HS lines were A (Ser) (Figure 8b, Appendix A). A significant DI difference was found between the two contrast haplotypes (using data of three experiments: Spring in 2014, autumn in 2014, and autumn in 2015) (Figure 8c). The expression of Csa5G606470 was comparable in Vlaspik and PI 197088 (Figure 8d).

For the locus *dmG4.1*, the candidate region (Chr.4: 5250–5350 kb) was analyzed by pairwise LD correlations leading to a region from 5,250,000 to 5,310,453 bp (Figure 9a). Four candidate genes were located in this region (Figure 9b). Only for the *Csa4G064680* gene, 10 SNPs showed a significant effect on DIs between haplotypes of the defined HR and HS lines (data of autumn of 2015). Of the ten SNPs, two were located in the CDS (Figure 9c, red letter), resulting in amino acid variation (Figure 9c,d). In the HR and HS lines, 20 out of the 22 HS lines carried the GATTATCTAG haplotype, while 6 out of the 18 HR lines had the CGCACATCGC haplotype (Appendix A). The expression of *Csa4G064680* was higher and upregulated starting from 2dpi in the resistant genotype PI 197088 (Figure 9e).

For the locus *dmG4.3*, no SNPs with a significant effect could be identified that colocalize with annotated genes in the analyzed region. The recently cloned *CsAAP2A* gene was located 200 kb away from the peak *dmG4.3.* Unfortunately, the mutant allele described in Berg et al. [12] could not explain the DM resistance of *dmG4.*3, since this mutation was not present in the CG lines (a subset of the resequenced 115 cucumber genome [13]) [12]. Further, the *CsLRK10L2* gene underlying the resistance conferred by the DM4.1 QTL in PI 197088 [11] was located 1Mb away from the peak *dmG4.2*. From the 32 CG lines that were identified carrying the PI 197088 allele [11], DI data were available for 27 lines (Appendix A). Surprisingly, a number of the 27 lines were highly susceptible, such as CG10, CG17, CG23, and CG110. Further, no significant difference was found between the mean DI of these 27 lines and the rest CG lines not carrying the PI 197088 allele (Appendix A).

For the locus *dmG7.2,* no SNPs with significant effect on DI could be identified.

## 3. Discussion

Many studies have shown that the DM resistance in cucumber is recessively inherited and polygenic [16,17,18,21,22,25,26,27,28,29]. Examples for the reported recessive genes include one recessive gene that is linked with the gene *D* for dull green fruit skin color [25] and the two recently cloned genes, *dm-1* and the *aap2a* gene. The *dm-1* gene derived from PI 197087 is a mutant allele of the *Staygreen* (*Sgr*) gene [8,9]. The *aap2a* gene in PI 197088 carries mutations in the *CsAAP2A* gene [11]. There are many examples including the current study that have reported polygenic DM resistance in cucumber and identified QTLs each with a major or minor effect [15,16,17,18,19,20,21,22,29,30,31,32,33]. QTLs for DM resistance can be found in all the seven cucumber chromosomes (Figure 4), suggesting a complex genetic architecture of DM resistance in cucumber.

In this study, phenotypic data on DI resistance were collected from natural infection in the fields. Many factors, such as the uniformity of infection, environmental conditions of different years/seasons, and pathogen population in the field, may influence the disease score. This is clearly reflected in our data by the mild infection in the spring of 2014 (2014S) and uneven distribution of the infection in the spring of 2016 (2016S). Fortunately, we could combine the DI data of different experiments, which helped prevent false positive evaluation on DM resistance. For example, a group of CG lines were scored HS in 2016S, in contrast to a lower DI scored in 2014S and the autumn of 2015 (2015A) (Figure 1). Therefore, we considered them susceptible to DM. Still, we could not rule out the possibility that the HR lines selected in this study may be due to an escape of infection; thus, they might be susceptible. Nevertheless, we succeeded in identifying 18 genetic loci contributing to DM resistance. Further, 16 of the 18 genetic loci colocalized with QTLs reported in previous studies (Figure 4), which was very supportive to our GWAS results. Thus, we conclude that the CG germplasm is rich in DM resistance and offers valuable donors for breeding cucumber with resistance to DM. However, it is worthwhile to note that of the prevailing DM isolates, many change from one season to the other, and the DM resistance found in the CG lines might be race-specific [34]. This could be one of the explanations for why DI data of 2016 did not have good correlations with the other three experiments. In order to confirm the resistance of these CG lines, an artificial DM inoculation with individual isolates under control conditions is recommended.

In cucumber, three genes, *Sgr* [9],* CsLRK10L2* [11], and *CsAAP2A* [12] for DM resistance have been cloned. Unfortunately, none of the three cloned genes were re-identified in this study. The *Sgr* gene is located on chromosome 5 in the region where none of our loci were located (Figure 4). The donor of the *sgr* mutant is Gy14 that was included in this study with a code of CG47. This line showed intermediate resistance and was not selected in the HR category. Possibly, DM race(s) in the fields of this study could overcome (partially) the resistance conferred by the *sgr* mutant. As for the *CsAAP2A* gene, Berg et al. [12] found an insertion of the CUMULE transposon in the mutant allele of the DM resistant accession PI 197088, leading to a very low expression of *CsAAP2A* in PI 197088 compared to the susceptible genotype. Unfortunately, the *csaap2a* mutant allele was not present in the core collection of sequenced 115 cucumber accessions [13]. Therefore, mutation in the *CsAAP2A* gene could not explain the effect of *dmG4.3* in this study. In addition, our results also showed that the *CsLRK10L2* gene, which is potentially the casual gene for subQTL DM4.1.2 in PI 197088 [11], could not explain the *dmG4.2* effect on DM resistance. It was surprising to see that the average DI of the CG lines carrying the PI 197088 allele of the *CsLRK10L2* gene was even more susceptible than the CG lines without this allele (Appendix A). Meanwhile, it is known that this subQTL, contributing to the restriction of DM sporulation, was only one of three subQTLs of the locus DM4.1 identified in PI 197088 [11]. The effect of this subQTL might be too small to be detected in this study. Alternatively, it might be that the isolates in the fields could be more aggressive overcoming the resistance conferred by this subQTL.

Several candidate genes were predicted as potential causal genes underlying the stable and novel loci, including *Csa1G575030* for *dmG1.4*, *Csa2G060360* for *dmG2.1*, *Csa4G064680* for *dmG4.1*, *Csa5G606470* for *dmG5.2*, and *Csa7G004020* for *dmG7.1*. For the locus *dmG5.*2, there was an SNP in the CDS of the *Csa5G606470* gene and this SNP variation resulted in amino acid changes. This candidate gene was also identified in the study of Zhang et al. [18] based on BSA-seq analysis. The *Csa5G606470* gene is homologous to the *Arabidopsis* gene *AT1G13960,* which was shown to be responsive to salicylic acid and a negative regulator of the defense response to bacterium [35]. In addition, *Arabidopsis* homologues of other candidate genes were shown to be related with plant defense responses. For example, the homologue gene of *Csa2G059830* in *Arabidopsis* was involved in regulation of defense response [36]. The *Arabidopsis* gene homologous to *Csa7G002520* was involved in the lignan biosynthetic process [37]. It has been shown that the lignans from *Myristica fragrans* have antifungal activity that suppressed the development of rice blast and wheat leaf rust [38].

To further study the predicted candidate genes, fine mapping would be the next step for a detailed genetic study by using a population where DM resistance segregates, as this was done in many other studies listed in Table 1. A QTL interval usually contains many genes, and genetic mapping will help to narrow down the QTL interval. This has been demonstrated in the study of Berg et al. [11], wherein three subQTLs were clustered in one QTL locus. Each subQTL contributes to DM resistance by limiting different infection processes of DM including chlorosis, sporulation, and DM-induced necrosis [11]. By developing and using near-isogenic lines, these subQTLs could be differentiated and causal genes underlying two subQTLs have been cloned [11,12]. Additionally, it may be helpful to check the gene expression of the HR and HS CG lines with contrast haplotypes upon DM infection. In this study, we used the available transcriptomic data (http://cucurbitgenomics.org/rnaseq/cu/19) of PI 197088 and Vlaspik as an indication. It is intriguing that the *Csa1G575010* gene located in the locus of *dmG1.4* was DM-induced in the resistant genotype PI 197088 in contrast to almost no detected expression in the susceptible genotype of Vlaspik (Figure 7e). This might indicate that the expression of this gene might play a role in DM resistance in PI 197088. For a final verification of the candidate genes, overexpression or knock-out the gene would be ideal.

## 4. Materials and Methods

### 4.1. Plant Materials

The 97 CG lines of cucumber were provided by the cucumber research group in the Institute of Vegetables and Flowers, Chinese Academy of Agricultural Sciences, China. This CG population was selected from more than 3000 germplasms worldwide and resequenced [13]. The code, origin, and ecotypes of these 97 CG lines are given in Appendix A. Sequencing data was on the cucumber genome website (http://www.icugi.org/cgi-bin/ICuGI/index.cgi).

### 4.2. Investigation of Disease Index of the Core Germplasm Resistance

All materials were grown in four experiments in the greenhouse of Nankou farm (40°13′ N, 116°09′ E) and Shunyi farm (40°15′ N, 116°83′ E) in Beijing in 2014 (spring and autumn), 2015 (autumn), and 2016 (spring). For each experiment, three replicates were applied and 6 plants per line were used in each replicate. The CG lines were randomized in each replicate (block). Under natural DM infection, symptoms appeared at about five weeks after sowing and scored three times once per week starting from about six weeks after sowing.

Disease rating scale was as follows: 0: No symptom; 1: ≤1/10 of all leaves with downy mildew spots; 3: 1/10~1/4 of all leaves with downy mildew spots; 5: 1/4~1/2 of all leaves with downy mildew spots; 7: 1/2~3/4 of all leaves with downy mildew spots; 9: ≥3/4 of all leaves with downy mildew spots, or the whole leaf dead [15]. Disease index (DI) was calculated as DI = 100 × ∑[number of plants with disease rating × disease rating]/[total number of plants × highest disease rating] [15]. All statistics for significant differences among different materials were performed using two-way analysis of variance or a two-tailed, two-sample Student’s *t*-test.

### 4.3. Genetic Diversity of DM Resistance in Germplasm

To cluster the CG lines based on their responses to DM infection, a phylogenetic tree was constructed using SAS 9.0 based on the average DI of each CG line in four experiments and presented in a heatmap [39].

### 4.4. Genome-Wide Association Analyses of DM Resistance

FastLMM was used for genome-wide association analyses, with an estimated relatedness matrix as the covariate, and the genome-wide minimal *p*-value was recorded. All 97 CG lines were included in GWAS. The 5% lowest minimal *p-*values were the threshold for genome-wide significance. The Manhattan map for GWAS was generated using *R* package CMplot [40]. SNP data used for the association analysis were from Qi et al. [13].

### 4.5. Linkage Disequilibrium Analysis

The software Plink [41] was used to calculate the LD coefficient (r^2^) between pairwise high-quality SNPs. the parameters were set as: ‘--r2 --ld-window 999999 --ld-window-kb 1000 --ld-window-r2 0′, and the results were used to estimate LD decay.

### 4.6. Candidate Gene Analysis

For each locus, the 50 kb around peak SNPs was considered as the region carrying candidate genes for DM resistance. The physical distance was based on the cucurbit genomics (http://cucurbitgenomics.org/). Using the LDblock program, the candidate genes were predicted by the annotation of homologous gene in *Arabidopsis* and the SNP variation. The QTL effect was calculated with the R/qtl package with the multiple-QTL model (MQM) [42,43].

## 5. Conclusions

In the cucumber CG germplasm, DM resistance was identified, and it is associated with 18 genetic loci, of which 16 were located in chromosomal regions where QTLs have been reported in previous studies. The genetic resistance in CG germplasm could be explained by the previously cloned DM genes including *Sgr, CsLRK10L2*, and *CsAAP2A*, suggesting that novel genes are present on the CG germplasm that contribute to the DM resistance. Based on the annotation of homologous genes in *Arabidopsis* and pairwise LD correlations, five genes were predicted to be the potential candidate genes underlying the resistance of four loci, *dmG1.4, dmG2.1, dmG4,1, dmG5.2*, and *dmG7.1*. This study shows that the CG germplasm is a very valuable resource carrying known and novel QTLs for DM resistance, which can be explored in breeding cucumber with resistance to DM.

## Figures and Tables

**Figure 1 plants-09-01659-f001:**
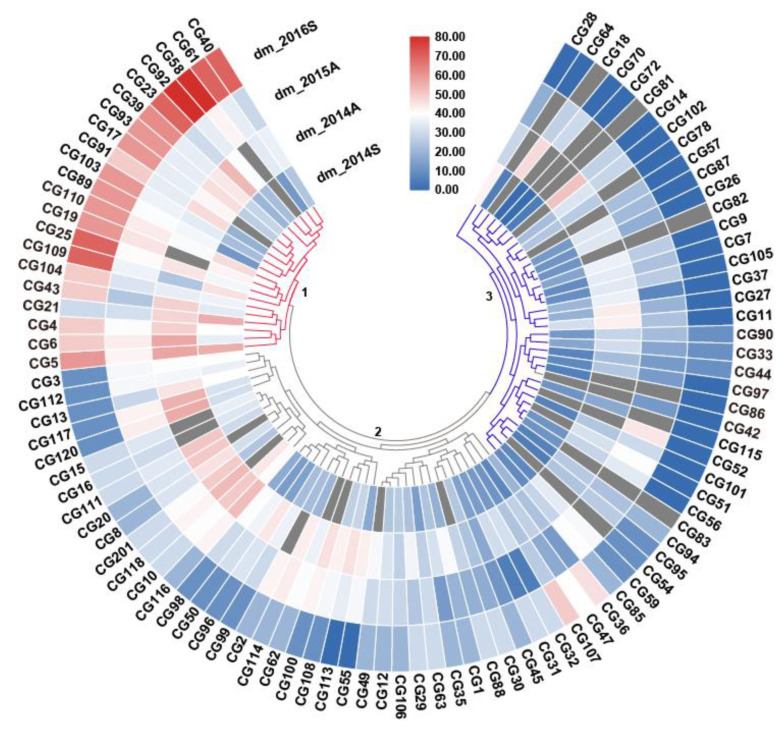
The heatmap depicting the phenotypic distribution of disease index (DI) of downy mildew (DM) resistance in four environments. The three clusters of the core germplasm (CG) lines are numbered with 1 to 3, and data of mean DI per CG line in the four experiments were used for this clustering analysis. The numbers next to the color key represent DI values. Blue means resistant and red susceptible. The intensity of the color indicates the level of resistance/susceptibility.

**Figure 2 plants-09-01659-f002:**
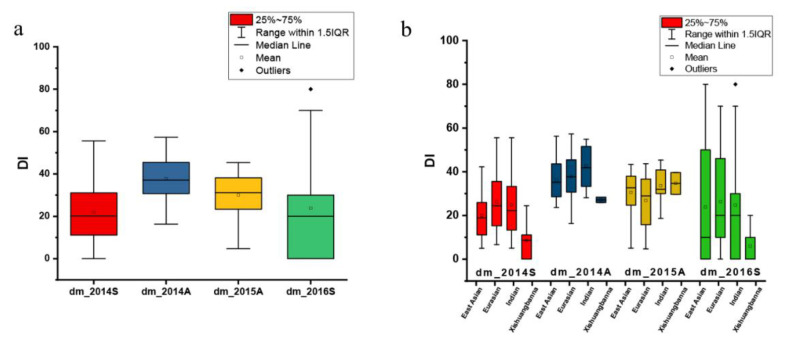
Box plots depicting the phenotypic distribution of disease index (DI) of downy mildew (DM) resistance in four experiments and among different ecotypes. (**a**) Phenotypic distribution in four experiments; (**b**) DI distribution in four ecotypes.

**Figure 3 plants-09-01659-f003:**
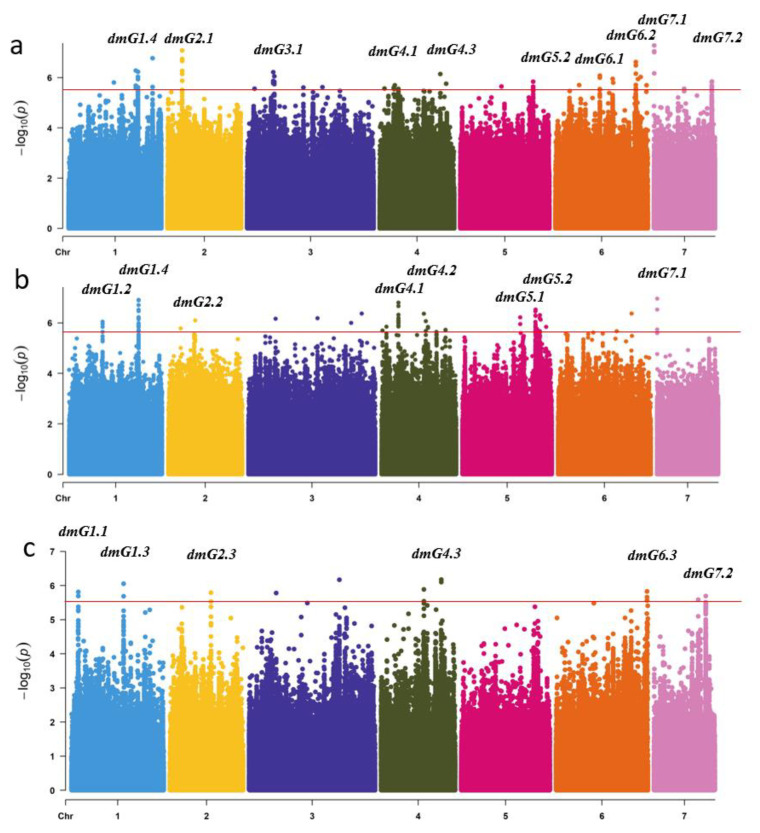
Genome-wide association analysis (GWAS) Manhattan plots of downy mildew (DM) resistance in three experiments, and the distribution of 97 cucumber CG lines in different continents based on DI of DM resistance. Bonferroni significance threshold of GWAS (*p*  <  3.15  ×  10^− 6^). (**a**) dm_2014A; (**b**) dm_2014S; (**c**) dm_2015A.

**Figure 4 plants-09-01659-f004:**
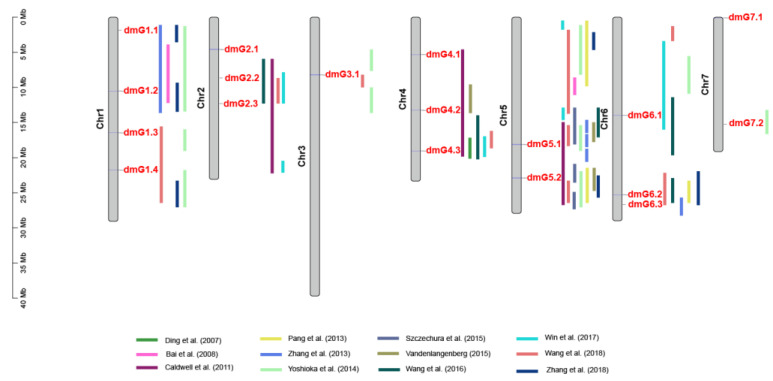
Locations of QTLs for downy mildew resistance in cucumber reported in this study and previous studies (2007–2018). The red font indicates the loci of this study and their exact SNP locations can be found in Appendix A.

**Figure 5 plants-09-01659-f005:**
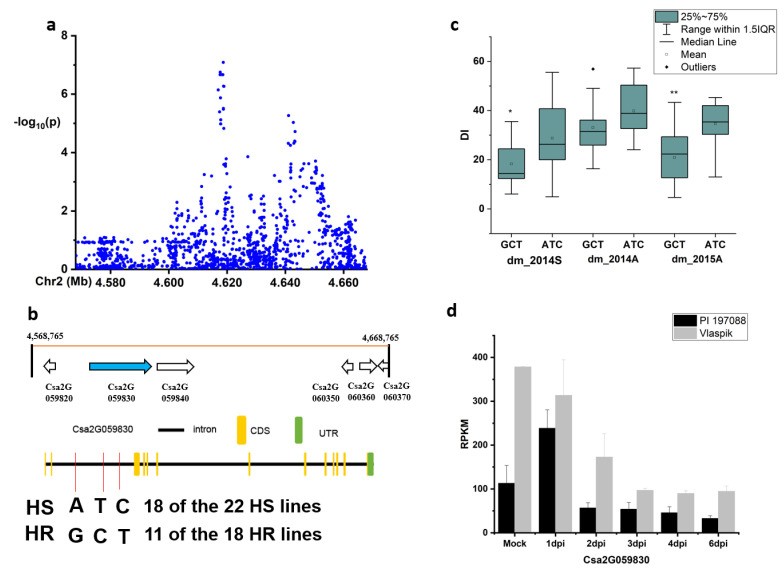
Identification of the causal gene for the locus *dmG2.*1. (**a**) Local Manhattan plot surrounding the peak of *dmG2.*1; (**b**) six genes were predicted in the 50-kb region around the peak *dmG2.1* and SNP variation of the candidate gene *Csa2G059830* between the highly susceptible (HS) and highly resistant (HR) lines (see Appendix A); (**c**) disease index (DI) comparison between haplotypes of the HS and HR lines (see Appendix A); (**d**) RPKM (Reads Per Kilobase Million) of candidate gene *Csa2G059830* based on the transcriptome of PRJNA285071 (data obtained from http://cucurbitgenomics.org/rnaseq/cu/19) [24]. * and ** indicate significance at *p* < 0.05 and *p* < 0.01, respectively.

**Figure 6 plants-09-01659-f006:**
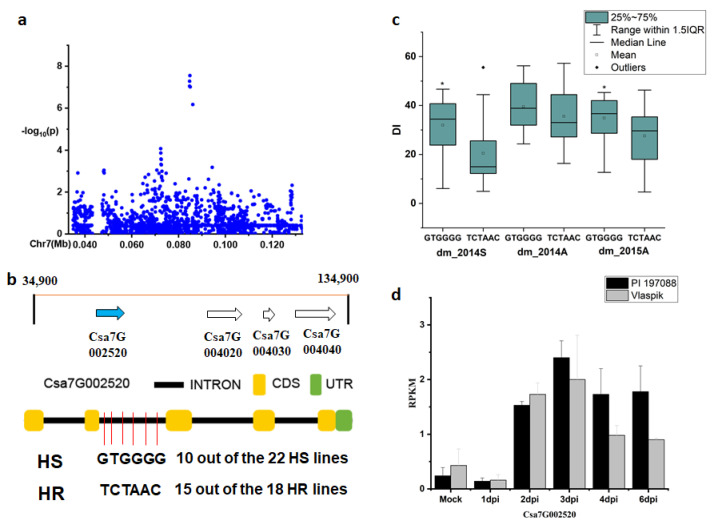
Identification of the causal gene for the locus *dmG7.1*. (**a**) Local Manhattan plot surrounding the peak of *dmG7.1*; (**b**) four genes predicted in the 50-kb region around the peak *dmG2.1* and SNP variation of the candidate gene *Csa7G002520* between the highly susceptible (HS) and highly resistant (HR) lines (see Appendix A); (**c**) disease index (DI) comparison between haplotypes of the HS and HR lines (see Appendix A); (**d**) RPKM of candidate gene *Csa7G002520* based on the transcriptome of PRJNA285071 (data obtained from http://cucurbitgenomics.org/rnaseq/cu/19) [24]. * indicates significance at *p* < 0.05.

**Figure 7 plants-09-01659-f007:**
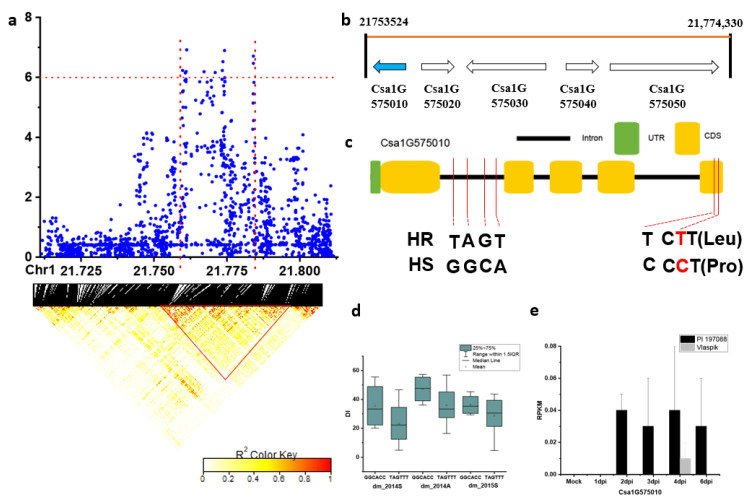
Identification of the causal gene for the locus *dmG1.4*. (**a**) Local Manhattan plot (top) and LD heatmap (bottom) surrounding the peak of *dmG1.4*; (**b**) five genes were predicted in the LD block region; (**c**) SNP variation of the candidate gene *Csa1G575010* between the highly susceptible (HS) and highly resistant (HR) lines (see Appendix A). The SNP in red color is located in the CDS resulting in amino acid variation. (**d**) Disease index (DI) comparison between haplotypes of the HS and HR lines (see Appendix A); (**e**) RPKM of the candidate gene *Csa1G575010* based on the transcriptome of PRJNA285071 (data obtained from http://cucurbitgenomics.org/rnaseq/cu/19) [24].

**Figure 8 plants-09-01659-f008:**
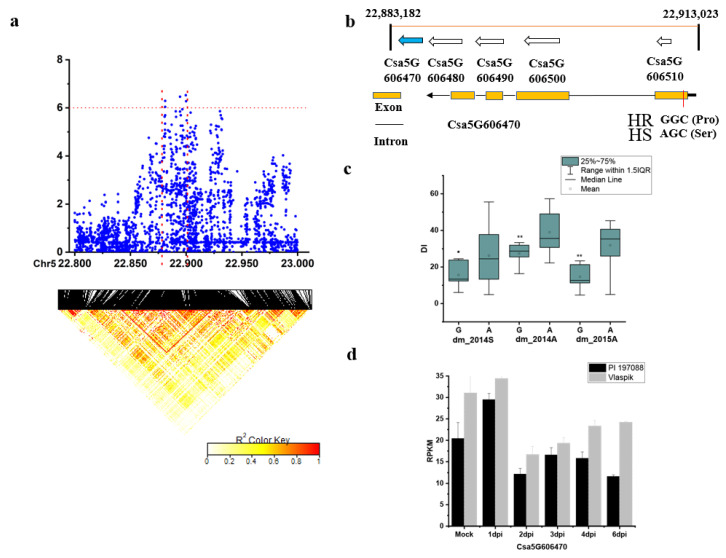
Identification of the causal gene for the locus *dmG5.2*. (**a**) Local Manhattan plot (top) and LD heatmap (bottom) surrounding the peak of *dmG5.2*; (**b**) five genes predicted in the LD block region and SNP variation of the candidate gene *Csa5G606470* between the highly susceptible (HS) and highly resistant (HR) lines (see Appendix A); (**c**) disease index (DI) comparison between haplotypes of the HS and HR lines (see Appendix A); (**d**) RPKM of the candidate gene *Csa5G606470* based on the transcriptome of PRJNA285071 (data obtained from http://cucurbitgenomics.org/rnaseq/cu/19) [24]. * and ** indicate significance at *p* < 0.05 and *p* < 0.01, respectively.

**Figure 9 plants-09-01659-f009:**
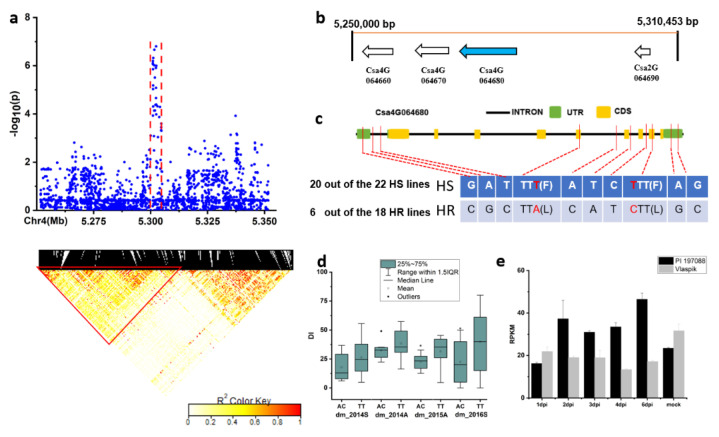
Identification of the causal gene for the locus *dmG4.1*. (**a**) Local Manhattan plot (top) and LD heatmap (bottom) surrounding the peak of *dmG4.1*; (**b**) four genes predicted in the LD block region. (**c**) SNP variation of the candidate gene *Csa4G064680* between the highly susceptible (HS) and highly resistant (HR) lines (see Appendix A). The two SNPs in red color are located in the CDS, resulting in amino acid variation. (**d**) Disease index (DI) comparison between haplotypes of the HS and HR lines (see Appendix A); (**e**) RPKM of the candidate gene *Csa4G064680* based on the transcriptome of PRJNA285071 (data obtained from http://cucurbitgenomics.org/rnaseq/cu/19) [24]. * indicates significance at *p* < 0.05.

**Table 1 plants-09-01659-t001:** Overview of mapped and cloned genes/quantitative trait loci (QTLs) for downy mildew resistance in cucumber.

Resistance Donor	Mapping Population	Gene/QTL	Chromo- Some	Causal Gene	Reference
PR40	-	*-*	-	*-*	Jenkins J.M., (1946) [5]
Indian accession PI 605996	-	*-*	-	*-*	Call et al., (2012) [14]
Indian accession PI 197087	RILs of Gy14 (R, derived from PI 197087) × 9930 and F_2:3_ families of WTF23 (R, derived from PI 197087) × True Lemon	*dm-1*	5	*Sgr*	Wang et al., (2019) [9]
Chinese long cucumber hybrid, Yuanfeng	F_2_ and F_2:3_ families of Inbred line K8 (R) × K18 (S)	*dm1.1*, *dm5.1*, *dm5.2*, *dm5.3, dm6.1*	1, 5, 6	-	Zhang et al., (2013) [15]; Bai et al., (2008) [16]
*Cucumis hystrix*	F_2_ population of IL52 (R) × Changcunmici (S)	*DM_5.1, DM_5.2*	5	-	Pang et al., (2013) [17]; Zhang et al., (2018) [18]
Pakistan accession PI 330628	F_2:3_ families of WI7120 (derived from PI 330628 (R)) × Chinses cultivar 9930 (S)	*dm2.1*, *dm4.1*, *dm5.1*, *dm6.1*	2, 4, 5, 6	-	Wang et al., (2016) [19]
Hybrid Malini	F_2:3_ families of TH118FLM (derived from Malini (R)) × WMEJ (S)	*dm2.2*, *dm4.1*, *dm5.1*, *dm5.2, dm6.1*	2, 4, 5, 6	-	Win et al., (2017) [20]
Indian accession PI 197088 & Japanese cultivar Santou	RILs of CS-PMR1 (derived from PI 197088, R) × Santou (intermediate S)	7 QTLs from CS-PMR1 3 QTLs from Santou	1, 3, 5 1, 6, 7	-	Yoshioka et al., (2014) [21]
Indian accession PI 197088	F_2:3_ families of PI 197088 (R) × Coolgree (S)	11 QTLs of which *dm5.1*, *dm5.2,* and *dm5.3* are major QTLs	All chromosomes except 7	-	Wang et al., (2018) [22]
Indian accession PI 197088	F_2_ and F_2:3_ families of PI 197088 × Changcunmici (S)	*dm1.1, dm3.1, dm4.1, dm5.1, dm5.2*	1, 3, 4, 5	-	Li et al., (2018) [23]
Indian accession PI 197088	NILs and F_2:3_ families of PI 197088 (R) × HS279 (S)	*DM4.1* and its derived subQTLs, *DM4.1.2* and *DM4.1.3*	4	*CsLRK10L2* *& CsAAP2A*	Berg J., (2020) [11]

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
