# Peer review of "Identification of Novel Loci and Candidate Genes for Cucumber Downy Mildew Resistance Using GWAS"

_plants, 2020, doi:10.3390/plants9121659_

Round 1

Reviewer 1 Report

In “Identification of novel loci and candidate genes for cucumber downy mildew resistance using GWAS”, Liu and colleagues present their mapping results for disease in cucumber. The mapped similar known loci from previous mapping efforts as well as identifying additional loci.

Overall the manuscript is missing detail and citations, calling into question the methods used to complete the study.

The introduction is missing detail and needs general revisions for grammar/spelling. The authors list many mapping studies in cucumber, but there is no integration or overall statements explaining general trends. What do those studies tell you as a whole? Further, I assume all the studies were mapping for DM resistance, but that isn’t clear. Later in the introduction, the authors state that GWAS are effective and list a few studies. Thousands of GWAS have been performed. Why are they citing this subset? It seems arbitrary and not particularly useful for the reader.

In the methods, it would be useful for the authors to share some history of DM at the study sites since they are examining local acquisition of DM. That leads to the major problem of the study. How do we know that each plant was exposed to DM? If DM were applied to the plants, why isn’t that discussed? How was the application performed What kind of experimental design was used (blocks)? If this fault is not addressed properly, this manuscript is not valid for publication.

Supplementary Figure 1 was not supplied in the submitted documents.

Please demonstrate the similarities between years 1 and 2 of the study using a scatterplot and describing the correlation in the text. The current display is appealing visually but is not useful from an informational perspective. It is very difficult to see if the pattern of similarity between years.

The examination of LD blocks varies from using 50 kb as an arbitrary distance to using LD blocks. Why did the authors switch between approaches?

I cannot read the text for the RPKM values in the results figures.

Reviewer 2 Report

Nice study; however, authors need to show how these genes are involved in DM resistance? Grammatical errors need to be corrected throughout the manuscript. I have listed only a few here.

Line# 29-30: “Although there are……..few genes been identified? Rewrite this segment.

Line# 32: “DM was occurred in many species…” ?????

Line# 34-35: These lines need to be rewritten.

Line# 37: The QTLs on Chr5……

Line# 38-43: Rewrite these sentences.

Line# 58: What these core germplasm are?

Line# 69-70: Spring and autumn have different moisture contents. Why you did not repeat in spring or autumn?

The manuscript needs many changes:

1. It needs an overhaul in terms of grammer. There are so many errors and I have pointed only a few of them.

2. Authors need to show that how the newly reported genes are involved in DM resistance?

3. Testing for DM resistance needs to be repeated for atleast one more season, either spring or fall.

These three comments are major comments, which need to be addressed.

Reviewer 3 Report

In the Introduction the authors display poor knowledge in the Oomycete world.

They claim (citations 5 and 6 ) downy mildew in wheat, which was never heard of...!!

In citation 5  they provide NO reference (probably a wrong citation).

In citation 6 the paper deals with powdery mildew.

I can hardly trust the results of authors that do not distinguish between downy mildew and powdery mildew. 

Round 2

Reviewer 1 Report

The revised manuscript is greatly improved. I appreciate the effort to make the paper clearer, and more interesting.

I believe only minor revisions/clarifications are necessary for publication. The most important of these is that I am very confused about how locus dmG4.3 was mapped if there were no SNPs of significant effect. How can you map a locus if there are no significant SNPs? Do you mean no SNPs colocalize with genes? Or do you mean that the DI in the HS and HR groups never varied at any of the mapped SNPs?

The methods require some additional clarifications. First, were only the HR and HS lines used in GWAS, or was the whole set of field trial lines used? What did you do about lines that were not included in all four field trials--were these lines used in GWAS with missing data?  Second, the dendrogram showing the relationships among the lines is referred to as a phylogenetic tree and the section in the methods is called Genetic diversity of DM resistance in germplasm. As far as I can tell, the tree was built with the DI phenotype, not the genotypes. Please clarify or correct the methods and figure 1 to reflect the accurate input data used. Also please include an indication on the color key on Figure 1 to explain the red and blue color is referring to DI.

There are data entry errors in Supp File 1, column dm_2014A

In Fig. 5-9, when comparing the haplotypes, can you clarify in the text (approximately line 101) that you are showing the average across 4 (or is it 3?) experiments plotted for each line?

In figure 5b, the labeling indicates that SNPs are found in the intron, not the CDS. However, in the text the six SNPs are said to be found in the CDS (line 119). Is there an error in the figure or the text that needs to be corrected?

In line 133, I don’t know what is meant by “stable locus”. Does this mean mapped in multiple experiments or mapped in many different published studies?

Is there a predicted function for Csa1G575010 that could be included in lines 137-138?

In line 69, the authors use the term “selected” to indicate the lines that they define as HS or HR. They use the term “selected” again in 102. The term has so many alternative meanings that I suggest changing it to something like defined, characterized, or identified, so it doesn’t like these lines are undergoing some breeding selection by the researchers.

In the discussion, at line 225, I would to the opening sentence that none of the three cloned genes were re-identified in this study, pointing to new loci of interest (if I am understanding the paragraph correctly).

Author Response

Dear Reviewer,

Thanks for your comments. We hope that all the comments have been addressed adequately in the revised manuscript.

Comments:

The revised manuscript is greatly improved. I appreciate the effort to make the paper clearer, and more interesting.

I believe only minor revisions/clarifications are necessary for publication. The most important of these is that I am very confused about how locus dmG4.3 was mapped if there were no SNPs of significant effect. How can you map a locus if there are no significant SNPs? Do you mean no SNPs colocalize with genes? Or do you mean that the DI in the HS and HR groups never varied at any of the mapped SNPs?

Response: We mean that no SNPs colocalize with genes. We made modification accordingly (line 150).

The methods require some additional clarifications. First, were only the HR and HS lines used in GWAS, or was the whole set of field trial lines used? What did you do about lines that were not included in all four field trials--were these lines used in GWAS with missing data?  Second, the dendrogram showing the relationships among the lines is referred to as a phylogenetic tree and the section in the methods is called Genetic diversity of DM resistance in germplasm. As far as I can tell, the tree was built with the DI phenotype, not the genotypes. Please clarify or correct the methods and figure 1 to reflect the accurate input data used. Also please include an indication on the color key on Figure 1 to explain the red and blue color is referring to DI.

Response: (1) All the 97 CG lines were included in GWAS, this is added to line 264. (2) For clustering the CG lines, we indeed used phenotypic data of DI. We have made adaptions to lines 259. This information is also added to the figure 1 legend. (3) The color key of Figure 1 is explained in the figure legend.   

There are data entry errors in Supp File 1, column dm_2014A

Response: Sorry for this error and we added the data in column dm_2014A.

In Fig. 5-9, when comparing the haplotypes, can you clarify in the text (approximately line 101) that you are showing the average across 4 (or is it 3?) experiments plotted for each line?

Response: We added the required information in the text to clarify the data used for each figure (Fig. 5 to 9).  

In figure 5b, the labeling indicates that SNPs are found in the intron, not the CDS. However, in the text the six SNPs are said to be found in the CDS (line 119). Is there an error in the figure or the text that needs to be corrected?

Response: The figure is correct, and we have revised in the text.

In line 133, I don’t know what is meant by “stable locus”. Does this mean mapped in multiple experiments or mapped in many different published studies?

Response: We meant a locus mapped in multiple experiments and have added “detected in more than one experiment”in the text (lines 57-58).

Is there a predicted function for Csa1G575010 that could be included in lines 137-138?

Response: The Csa1G575010 gene encodes a Pectinesterase. We added the information to line 103.

In line 69, the authors use the term “selected” to indicate the lines that they define as HS or HR. They use the term “selected” again in 102. The term has so many alternative meanings that I suggest changing it to something like defined, characterized, or identified, so it doesn’t like these lines are undergoing some breeding selection by the researchers.

Response: Thanks for this comment. We have changed “selected” to “defined”(see lines 66, 85, 102 and line 142).

In the discussion, at line 225, I would to the opening sentence that none of the three cloned genes were re-identified in this study, pointing to new loci of interest (if I am understanding the paragraph correctly).

Response:  Yes, the three cloned genes were not re-identified in this study. We took this suggestion and added this sentence to line 199.

With kind regards,

Shengping Zhang (on behalf of all authors)

Reviewer 3 Report

Four experiments were conducted to assess the DI of the CG lines.

In one experiment (dm 2016s) the results did not correspond to the other 3 experiments and the authors decided to delete this experiment from analysis.

I assume that if another experiment would have been conducted the results might have been again different.

The reason is the prevailing isolate(s) of P. cubensis may change from one season to the other, and since resistance is isolate-dependent the conclusions regarding the OTLs for resistance might be wrong. I draw the attention of the authors  to a recent paper that deals with this problem:

  1. Chen, T., Katz, D., Ben Naim, Y. Ben Daniel, B.H. Hammer r., and Cohen, Y. 2020.   Isolate‐dependent inheritance of resistance against Pseudoperonospora cubensis in cucumber. Agronomy 2020, 10, 1086; doi:10.3390/agronomy10081086. On line 27.7.2020. 25 pp

Author Response

Dear reviewer,

Thanks for your comments. Based on your comments, we have revised our manuscript accordingly. We hope that all the  comments have been addressed adequately in the revised manuscript. 

Comments and Suggestions for Authors

Four experiments were conducted to assess the DI of the CG lines.

In one experiment (dm 2016s) the results did not correspond to the other 3 experiments and the authors decided to delete this experiment from analysis.

I assume that if another experiment would have been conducted the results might have been again different.

The reason is the prevailing isolate(s) of P. cubensis may change from one season to the other, and since resistance is isolate-dependent the conclusions regarding the OTLs for resistance might be wrong. I draw the attention of the authors to a recent paper that deals with this problem:

  1. Chen, T., Katz, D., Ben Naim, Y. Ben Daniel, B.H. Hammer r., and Cohen, Y. 2020.   Isolate‐dependent inheritance of resistance against Pseudoperonospora cubensis in cucumber. Agronomy 2020, 10, 1086; doi:10.3390/agronomy10081086. On line 27.7.2020. 25 pp

Response: Thanks for this helpful comment. This can be indeed one additional explanations and we added it by citing this paper (lines 191 to 195).   

With kind regards,

Shengping Zhang (on behalf of all authors)